# Odor-evoked category reactivation in human ventromedial prefrontal cortex during sleep promotes memory consolidation

Laura K Shanahan[1]*, Eva Gjorgieva[1], Ken A Paller[2], Thorsten Kahnt[1,2], Jay A Gottfried[1,3,4]

[1]Department of Neurology, Feinberg School of Medicine, Northwestern University, Chicago, United States; [2]Department of Psychology, Weinberg College of Arts and Sciences, Northwestern University, Evanston, United States; [3]Department of Neurology, Perelman School of Medicine, University of Pennsylvania, Philadelphia, United States; [4]Department of Psychology, School of Arts and Sciences, University of Pennsylvania, Philadelphia, United States

*For correspondence:
laurashanahan2012@u.
northwestern.edu

Competing interests: The authors declare that no competing interests exist.

**Abstract** Slow-wave sleep is an optimal opportunity for memory consolidation: when encoding occurs in the presence of a sensory cue, delivery of that cue during sleep enhances retrieval of associated memories. Recent studies suggest that cues might promote consolidation by inducing neural reinstatement of cue-associated content during sleep, but direct evidence for such mechanisms is scant, and the relevant brain areas supporting these processes are poorly understood. Here, we address these gaps by combining a novel olfactory cueing paradigm with an object-location memory task and simultaneous EEG-fMRI recording in human subjects. Using pattern analysis of fMRI ensemble activity, we find that presentation of odor cues during sleep promotes reactivation of category-level information in ventromedial prefrontal cortex that significantly correlates with post-sleep memory performance. In identifying the potential mechanisms by which odor cues selectively modulate memory in the sleeping brain, these findings bring unique insights into elucidating how and what we remember.
DOI: https://doi.org/10.7554/eLife.39681.001

## Introduction

Only a small fraction of the events that are experienced during wakefulness are stamped into long-term memory. Understanding how memories are formed, and which memories are ultimately retained or forgotten, is a pivotal focus of neuroscience research. The process through which memories are stabilized and integrated for long-term storage, termed memory consolidation, is robustly enhanced during sleep (*Deak and Stickgold, 2010*; *Diekelmann and Born, 2010*). Although the neural mechanisms underlying sleep-based consolidation are far from clear, research increasingly implicates memory replay, whereby the same neural activity that occurs during memory encoding comes back online spontaneously during sleep to facilitate integration of the replayed memory into distributed cortical networks for long-term storage. Direct cellular-level evidence for replay comes from rodent studies (*Skaggs and McNaughton, 1996*; *Wilson and McNaughton, 1994*), while in humans, an indirect measure often termed 'reactivation' has also been demonstrated using fMRI, surface EEG, and intracranial EEG techniques (*Bergmann et al., 2012*; *Deuker et al., 2013*; *Peigneux et al., 2004*; *Schönauer et al., 2017*; *Zhang et al., 2018*).

**eLife digest** It may not always feel like it, but we encounter an incredible volume of new information every day. We experience so much that it is not feasible to remember every detail. The brain's process for reorganizing memories – keeping some secure and discarding others – is known as memory consolidation. There are ways of directing consolidation toward certain memories. One of them is to associate a memory – such as an arrangement of objects – with a particular smell. If this odor is then wafted at the person when they sleep, they are better at recalling the associated memory the next day.

The neural mechanisms in the brain that support this process are largely unknown. Researchers want to find out exactly how odor cues can alter brain activity while participants are asleep to allow for better recall on awakening.

Shanahan et al. used fMRI scans to see how an odor affects the sleeping brain. First, the participants learned the locations of several objects on a four by four grid – including animals, faces, buildings and tools – and then learned to associate each category with a different background odor. Then, the volunteers took a nighttime nap inside the MRI scanner, and were exposed to two of the odors in their sleep. The next morning, they better remembered the locations of the objects from the two categories associated with the odor cues delivered in sleep. Analyses of the brain scans revealed that the extent to which odors reactivated the category information in a part of the brain called the ventromedial prefrontal cortex was predictive of how successful memory recall was after sleep. This brain region is involved in retrieving old memories.

Memory disorders are an ever-increasing problem as the average life-span continues to rise. Reliable treatments to slow or prevent memory decline in patients with conditions such as Alzheimer's are still unavailable. Using odor cues during sleep could be one way to enhance memories in patients with memory loss and dementia, but also in healthy individuals.
DOI: https://doi.org/10.7554/eLife.39681.002

Intriguingly, it has been shown that external sensory cues, such as odors or sounds, can be presented during sleep to manipulate what information is preserved. After a sensory cue is presented during memory encoding in the wake state, re-presentation of that cue during sleep favors subsequent retrieval for the associated material (*Oudiette and Paller, 2013*; *Rasch et al., 2007*; *Rudoy et al., 2009*; *Schouten et al., 2017*; *Shanahan and Gottfried, 2017*; *Spiers and Bendor, 2014*). It follows that cue-evoked reactivation of content-specific information during sleep might promote these memory gains, but direct evidence for such mechanisms is highly limited (*Schouten et al., 2017*). Groundbreaking work in rodents showed that auditory cues induce within-sleep replay of spatial sequences in hippocampal place cells, although these effects were not accompanied by an index of learning (*Bendor and Wilson, 2012*). Recent human studies have tested the idea that patterns of EEG activity in sleep can be used to decode different forms of auditory-cued memories, including procedural sequence learning (*Belal et al., 2018*) and place versus object information (*Cairney et al., 2018*), with the latter study demonstrating a correlation with post-sleep memory performance. However, because the spatial resolution of surface EEG measures is poorly suited for pinpointing the involvement of specific brain regions and networks, there remains a critical gap in understanding how cued reactivation of specific memory content is induced in the sleeping brain, and whether content-specific reactivation holds relevance for behavior.

To address this important gap, we designed a novel olfactory fMRI paradigm optimized to investigate the functional links between odor-evoked memory reactivation and declarative memory consolidation. Here, subjects completed an object-location memory task, in which objects belonged to four different categories (animals, buildings, faces, tools) (*Figure 1*). In turn, each object category was associated with a different odor cue (e.g. banana, cedar, cinnamon, garlic). Critically, the use of an MRI-compatible EEG system enabled us to deliver these odor cues selectively during slow-wave sleep (SWS), such that odor-evoked patterns of fMRI activity in the sleeping brain could be compared to canonical category representations from a preceding wake scan. In this manner, we were able to identify brain regions where fMRI signatures of within-sleep reactivation would have a direct impact on the strength of memory recall, as a function of reactivation strength within subjects.

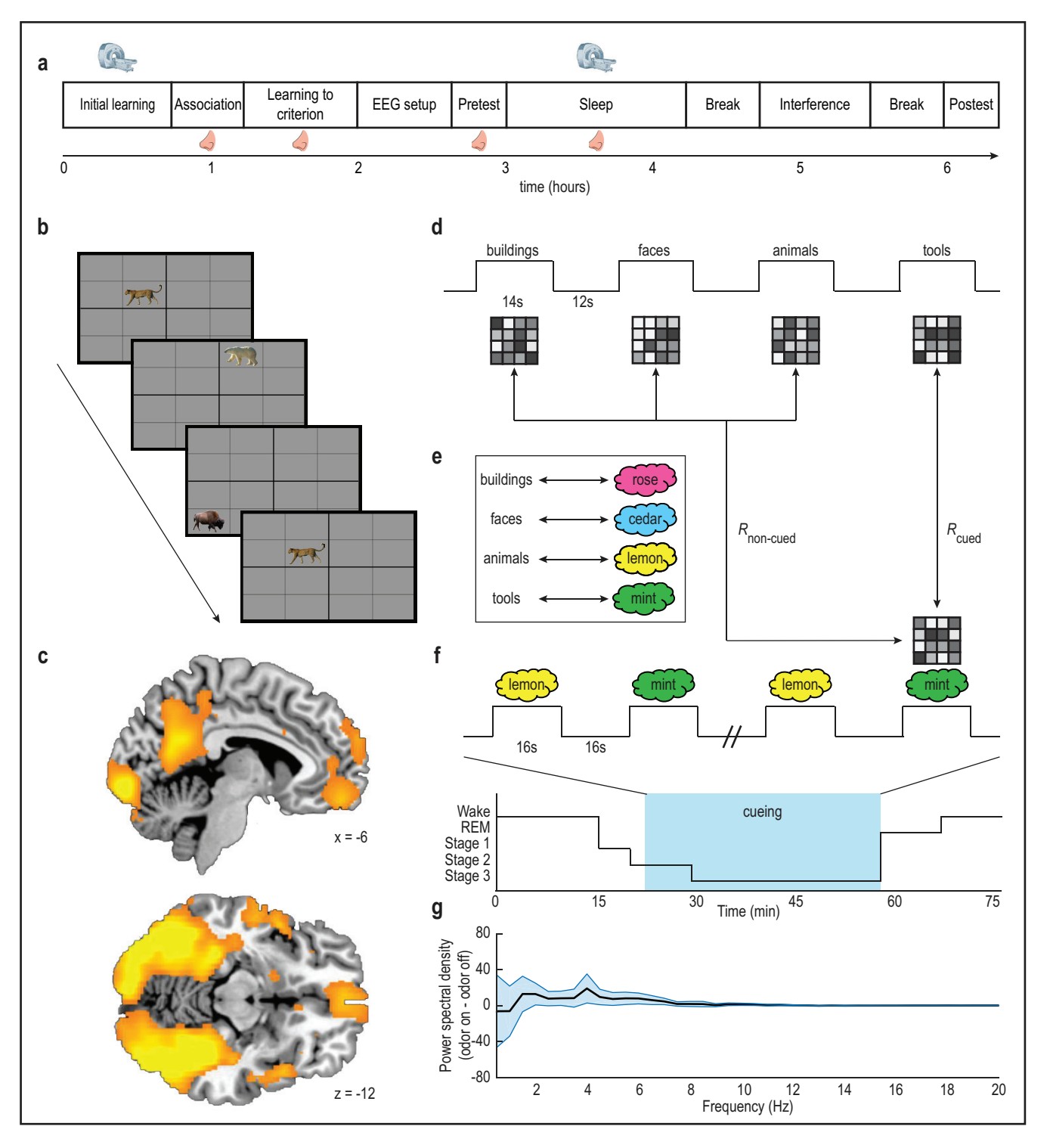

**Figure 1.** Olfactory cueing paradigm. (**a**) Experimental timeline. MRI scanner symbols indicate scanned study phases, and nose symbols indicate study phases during which odors were presented. (**b**) Initial learning task. Subjects learned spatial locations of objects from four categories (animals, buildings, faces, tools; 32 objects per category) during blocked-design fMRI scanning. Category-sensitive voxels were then identified for each subject by comparing visually evoked fMRI ensemble patterns of activity across runs, and calculating within-category versus between-category pattern correlations. This multivoxel pattern analysis was implemented using a whole-brain searchlight approach. (**c**) Category-sensitive brain regions included visual processing areas, and parts of parietal cortex and ventromedial prefrontal cortex. fMRI activity is displayed at p < 0.001 uncorrected, and images are overlaid on a canonical single-subject T1-weighted MRI scan. Voxels from this analysis were retained as an image mask for the main analysis. (**d-e**)

*Figure 1 continued on next page*

*Figure 1 continued*

During initial learning, category templates (depicted as 4-x-4 greyscale grids of voxels) were defined for each subject. Next, subjects learned to associate each object category with a unique odor (e.g. mint odor + tool images). (f) Subjects were placed into the fMRI scanner, and during NREM sleep stages 2 and 3, two of the four odor cues were presented in 16 s on/16s off blocks during fMRI scanning (i.e. during cueing, blue box) to cue content from their associated object categories. In this example, presentation of the mint odor in sleep would induce reactivation, such that its fMRI ensemble pattern would more closely resemble the tool category ensemble pattern defined in the wake state during initial learning. (g) There was no significant effect of odor presentation on power spectral density ($\mu V^2$/Hz) between 0.5 and 20 Hz (repeated-measures ANOVA, condition by frequency interaction: $F_{(39,17)}$ = 0.31, p = 1.00). Error bars depict mean ±SEM across subjects.

DOI: https://doi.org/10.7554/eLife.39681.003

The following source data and figure supplements are available for figure 1:

**Source data 1.** Relates to panel (g).
DOI: https://doi.org/10.7554/eLife.39681.007
**Figure supplement 1.** Brain regions demonstrating category selectivity during initial learning.
DOI: https://doi.org/10.7554/eLife.39681.004
**Figure supplement 2.** Differences in EEG spectral power between stage 3 and stage 1 sleep.
DOI: https://doi.org/10.7554/eLife.39681.005
**Figure supplement 2—source data 1.** Relates to entire figure.
DOI: https://doi.org/10.7554/eLife.39681.006

Specifically, we predicted that we would observe this relationship in task-relevant sensory brain areas (*Ji and Wilson, 2007*; *Rothschild et al., 2017*), and in memory retrieval networks in the hippocampus (*Bendor and Wilson, 2012*; *Peigneux et al., 2004*; *Rasch et al., 2007*) and prefrontal cortex (PFC) (*Bonnici and Maguire, 2018*; *Euston et al., 2012*; *Jin and Maren, 2015*; *Preston and Eichenbaum, 2013*; *Takashima et al., 2007*).

## Results

Thirty-two healthy human subjects (21 female; *mean age* = 25.25 years; *age range* = 19–37 years) participated in the study. Fourteen subjects were excluded because they did not receive the minimum number of odor cues during sleep (see Materials and methods), and consequently 18 of the 32 subjects were included in the analysis. The main experiment took approximately 6.5 hr to complete and consisted of several different task blocks (*Figure 1a*), as briefly outlined here. Human subjects first performed a visuospatial memory task, in which they learned the locations of objects from four categories during fMRI scanning. The purpose of this initial learning phase was to define fMRI ensemble representations (effectively, pattern templates) of each object category. Next, subjects learned to associate each of the four object categories with one of four distinct odors. Then, subjects were fitted with EEG caps and took a nap during fMRI scanning, during which time two of the odors were re-presented in sleep to selectively cue object representations from the associated categories. Memory for object locations was tested both before and after the sleep phase. Finally, the extent to which category-specific patterns of fMRI activity re-emerged in response to odors during within-sleep cueing (i.e. memory reactivation) was correlated with visuospatial memory performance on a subject-by-subject basis. To enhance the likelihood of observing sleep during fMRI scanning, the experiment took place at night so that the sleep phase was aligned approximately with each subject's habitual bedtime.

### Category-specific objects induce widespread discriminable patterns of fMRI activity during visuospatial learning

In an initial learning phase, subjects performed a visuospatial memory task, in which they learned the locations of objects from four categories: animals, buildings, faces, and tools. There were 32 objects per category, and object images were presented on a 4-x-4 spatial grid during fMRI scanning (*Figure 1b*). Subjects were instructed to memorize the location associated with each object, each of which appeared three times over the course of the scan. In line with previous studies demonstrating that visual category perception elicits unique ensemble patterns of fMRI activity (*Haxby et al., 2001*; *Norman et al., 2006*), a multivoxel pattern-based fMRI searchlight analysis revealed category specificity in widely distributed brain regions, including much of the visual pathway, and substantial parts

of parietal and prefrontal cortices (*Figure 1c*; *Figure 1—figure supplement 1*). fMRI data collected from the initial learning phase were then used to define multivoxel representations of stimuli belonging to each object category (*Figure 1d*), for use as reference templates to identify content-specific fMRI activity that might emerge in subsequent sleep. Importantly, these pattern templates were defined prior to the introduction of odor cues, to ensure that fMRI templates exclusively reflected visual category information.

## Odor cues bias memory consolidation toward their associated object categories

After completing the initial learning phase, subjects were moved from the scanner to a testing room, where they learned to associate four easily-distinguishable odors with each of the four object categories to criteria (*Figure 1e*). Based on individual ratings, each subject received a unique combination of four odors that were maximally discriminable (see Materials and methods), from a set of eight familiar odors (banana, cedar, cinnamon, garlic, lemon, mint, rose, vanilla). Following odor-category learning, subjects completed another learning session to reinforce odor-category associations and to learn object locations to criteria, and were then fitted with an MRI-compatible EEG electrode cap. In a subsequent memory pretest session, subjects were asked to recall object locations (without feedback). This test took place immediately before the fMRI sleep session, providing a memory baseline for comparison to post-sleep memory performance. Recall accuracy at pretest was robust (*Figure 2a*, first column), without significant differences between those categories that were later cued during sleep and those that were not ($t_{(17)}$ = 0.62, p = 0.54; *Figure 2—figure supplement 1*).

Subjects then returned to the scanner, where they were instructed to try to relax and fall asleep during approximately 75 min of fMRI scanning. During sleep stage 2 and SWS, two of the four odors were presented (16 s on/16 s off) via a Teflon tube secured beneath the nose, to cue two of the four object categories (*Figure 1f*; *Figure 1—figure supplement 2*). Selection of the odor cues in sleep was arranged to ensure that the specific categories designated for reinforcement were counterbalanced across subjects. Eighteen subjects reached criteria, in that during sleep they received a minimum of 14 presentations of each odor cue, and during debriefing they did not recall having smelled any odors during the scanning session. On average, these 18 subjects slept for 65.36 min, of which 26.50 min were spent in sleep stage 2, and 30.17 min were spent in SWS (*Table 1*). Analysis of the within-sleep EEG data did not reveal significant spectral differences between odor-on and odor-off periods (repeated-measures ANOVA, time by cue interaction: $F_{(39,17)}$ = 0.31, p = 1.00; *Figure 1g*), suggesting that odor delivery was not associated with physiological arousal.

Upon waking, but prior to the final post-sleep memory test, subjects were removed from the scanner and taken to a behavioral testing room, where they underwent a memory interference task to learn new grid locations for the same set of objects. This maneuver has been shown to provide added sensitivity to identifying effects of sleep on memory performance (*Ellenbogen et al., 2009*). Five minutes after completing the interference encoding task, subjects were asked to try to place the objects in their new locations. As predicted, overall recall accuracy on the interference task was lower compared to pre-sleep recall accuracy (*Figure 2a*, second column). Recall accuracy did not differ for cued compared to non-cued objects ($t_{(17)}$ = 0.04, p = 0.97). However, odor cues significantly influenced recall speed for the new object locations, whereby response times (RTs) for placing objects on the grid were slower for cued versus non-cued objects ($t_{(17)}$ = 2.76, p = 0.01; *Figure 2b*), probably reflecting increased competition between the new object locations and the original object locations that were reinforced by odor cues during sleep. This finding provides a first indication that odor cues presented in sleep had an impact on memory storage.

Finally, after a 30-min break, subjects were asked to recall the original object locations during a posttest session, for direct comparison to pretest performance. Posttest recall (*Figure 2a*, third column) was predictably lower than pretest recall ($t_{(17)}$ = 10.06, p < 0.001). Critically, subjects demonstrated less forgetting for cued object locations from pretest to posttest, when compared to non-cued object locations (Z = 1.70, p = 0.04; Wilcoxon signed-rank test; *Figure 2c*), with selective memory gains for 15/18 subjects (p = 0.003; Binomial test). These findings, as well as the RT data, confirm that delivery of odor cues during SWS selectively biased recall toward object categories previously paired with those odors in the wake state. Of note, although memory performance was significantly above chance during a follow-up memory test 1 week after the main experiment (*Figure 2a*, fourth

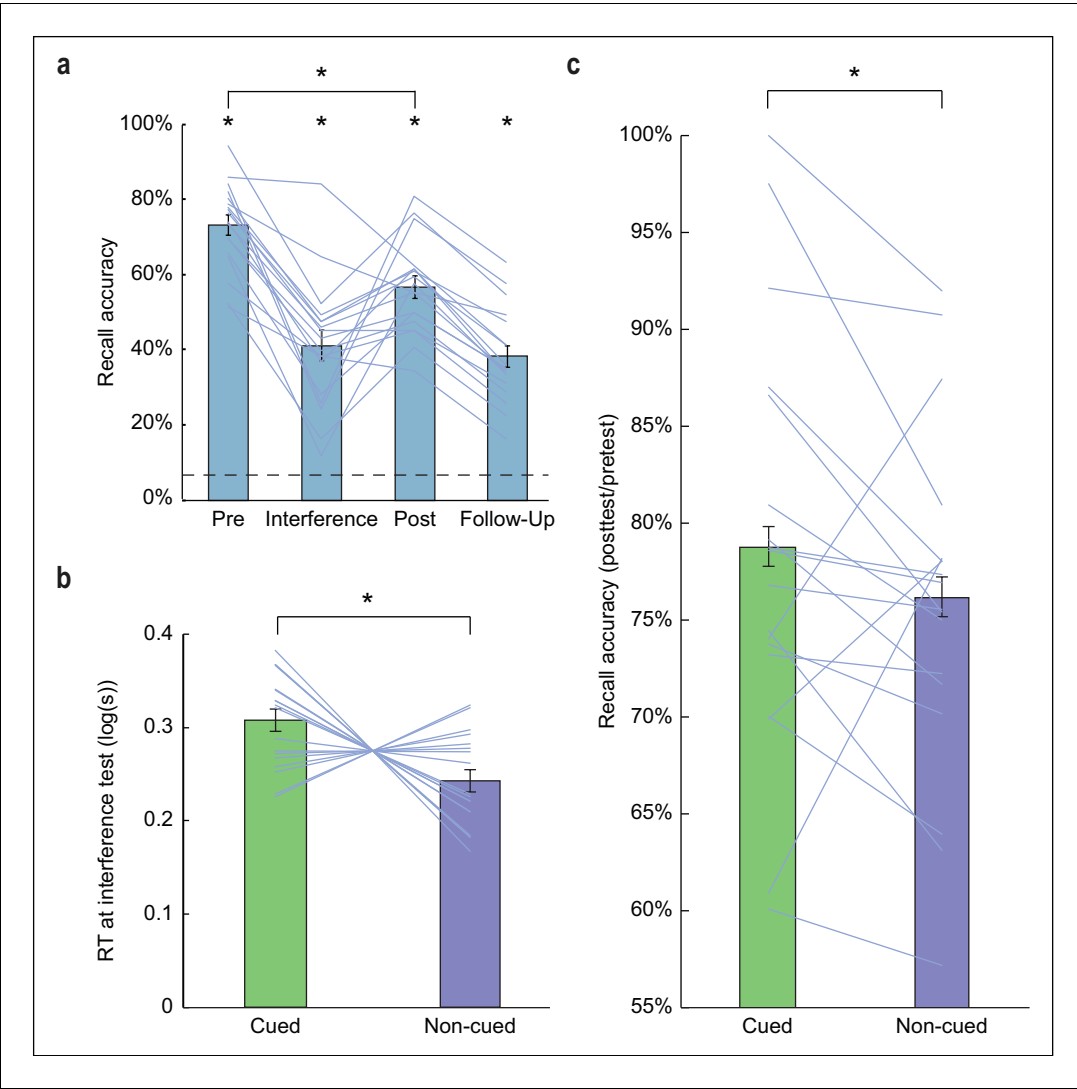

**Figure 2.** Behavioral results. (a) Subjects demonstrated robust visuospatial recall (chance = 6.25%; dashed line) at pretest (73.17% correct ±2.71% SEM, $t_{(17)}$ = 24.68, p < 0.001), interference test (41.11% correct ±4.02% SEM, $t_{(17)}$ = 8.66, p < 0.001), posttest (56.70% correct ±2.92% SEM, $t_{(17)}$ = 17.27, p < 0.001), and 1 week follow-up (38.24% correct ±2.92% SEM, $t_{(17)}$ = 10.96, p < 0.001). Performance declined significantly from pretest to posttest ($t_{(17)}$ = 10.06, p < 0.001). (b) During interference testing, subject RTs were significantly slower for objects belonging to categories that were cued in the preceding sleep period, versus non-cued categories. This suggests that memory for object categories that were cued during SWS was more resistant to interference. *p < 0.05, two-tailed *t*-test. (c) At posttest, recall accuracy (percentage of items recalled at posttest, compared to pretest baseline) was enhanced for cued object locations, compared to non-cued object locations (Z = 1.70, p = 0.04). *p < 0.05, one-tailed Wilcoxon signed-rank test. Error bars in (a) depict mean ±SEM, and error bars in (b–c) depict mean ±within subjects SEM.

DOI: https://doi.org/10.7554/eLife.39681.008

The following source data and figure supplements are available for figure 2:

**Source data 1.** Relates to panel (a).
DOI: https://doi.org/10.7554/eLife.39681.011
**Source data 2.** Relates to panel (b).
DOI: https://doi.org/10.7554/eLife.39681.012
**Source data 3.** Relates to panel (c).
DOI: https://doi.org/10.7554/eLife.39681.013
**Figure supplement 1.** Visuospatial recall accuracy at pretest.
DOI: https://doi.org/10.7554/eLife.39681.009

*Figure 2 continued*

**Figure supplement 1—source data 1.** Relates to entire figure.
DOI: https://doi.org/10.7554/eLife.39681.010

column), effects of odor cueing on memory performance were no longer evident (recall accuracy: $t_{(17)}$ = 0.17, p = 0.43; RT: $t_{(17)}$ = 0.46, p = 0.65).

## An index of odor-evoked category reactivation in the sleeping brain predicts post-sleep memory performance

Having established the selective influence of odor cueing on recall performance, we next examined the mechanisms underlying this memory-enhancing effect. Our central hypothesis was that if odor cues induce within-sleep reactivation of associated category information in medial temporal and prefrontal brain areas known to participate in memory consolidation and retrieval (*Bonnici and Maguire, 2018*; *Euston et al., 2012*; *Jin and Maren, 2015*; *Preston and Eichenbaum, 2013*; *Takashima et al., 2007*), then the degree of memory reactivation should predict subsequent memory performance in the wake state. To this end, we first correlated odor-evoked fMRI ensemble activity in sleep with each of the four category-specific fMRI template patterns from the initial learning phase of the task, and then compared the degree of pattern overlap between cued and non-cued category templates, yielding a measure of cue-specific reactivation strength for each subject (*Figure 1d,f*). By regressing this within-sleep reactivation measure onto changes in memory performance from pretest to posttest, we were able to determine whether memory recall varied with reactivation strength across subjects. Note, this analysis was restricted to regions that demonstrated category specificity during prior learning. In this way, we found that greater category reactivation in ventromedial PFC (vmPFC) was associated with increased recall accuracy at posttest for cued over non-cued objects ([−6, 46,−12], $t_{(16)}$ = 7.53, $p_{FWE}$ = 0.01, *Figure 3a*). These effects were robust across both odor cues, as within-sleep reactivation strength in vmPFC was significantly correlated with post-sleep recall when each cue was considered independently ($r_{1(16)}$ = 0.53, $p_1$ = 0.01; $r_2$ $_{(16)}$ = 0.48, $p_2$ = 0.02; *Figure 3b*). In a time-resolved analysis of stimulus-evoked activity during the sleep period, correlations between odor-evoked reactivation in vmPFC and recall accuracy increased at odor onset and persisted over several seconds, with a maximal effect size (*r* value) of 0.70 (90% confidence interval = 0.42 to 0.86), returning to baseline prior to odor offset (*Figure 3c*). Together these findings in vmPFC highlight the categorical and temporal specificity of odor-cued reactivation on memory retrieval (see *Figure 4* for additional analysis).

Because our paradigm fundamentally involves a visuospatial task, we also reasoned that within-sleep reactivation of cued content might manifest in visual associative brain regions (*Ji and Wilson, 2007*; *Rothschild et al., 2017*). Across subjects, recall accuracy for cued versus non-cued object categories scaled with the degree of odor-evoked category reactivation in posterior fusiform cortex

**Table 1.** Time spent in each sleep stage.
Offline sleep scoring revealed that 99.45% of odors were presented during stages 2 and 3 of sleep, and most cues (77.56%) were presented during stage 3 of sleep.

| Sleep stage | Time (min ±SEM) | Percentage (% ±SEM) |
| --- | --- | --- |
| Wake | 9.92 ±2.61 | 13.11 ±3.48 |
| Stage 1 | 8.69 ±1.82 | 11.39 ±2.34 |
| Stage 2 | 26.50 ±3.41 | 34.95 ±4.26 |
| Stage 3 | 30.17 ±4.12 | 40.54 ±5.56 |
| Total sleep time | 65.36 ±2.79 | 86.89 ±3.48 |
| Total time (wake + sleep) | 75.28 ±1.25 | 100 |

DOI: https://doi.org/10.7554/eLife.39681.029
The following source data is available for  Table 1:
**Source data 1.** Time spent in each sleep stage by subject.
DOI: https://doi.org/10.7554/eLife.39681.030

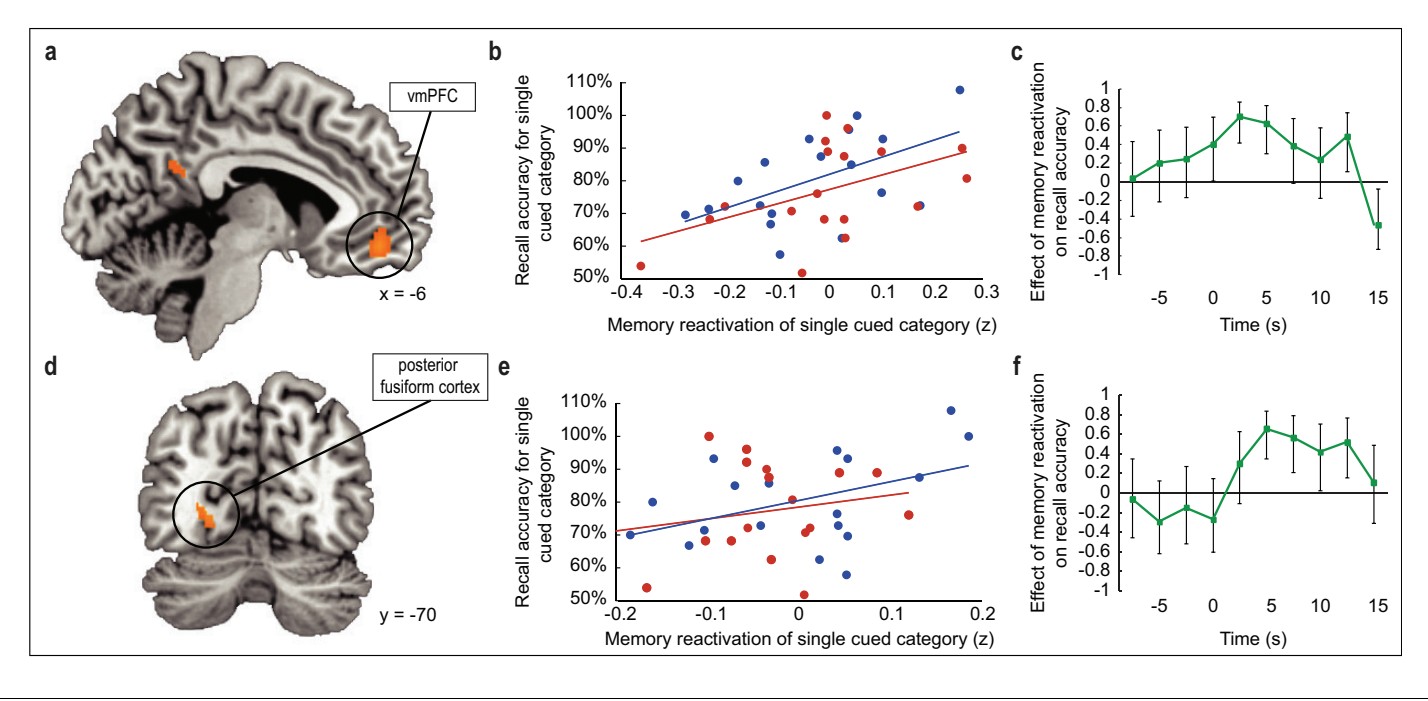

**Figure 3.** Relationship between odor-evoked memory reactivation in sleep and posttest memory retention. (**a,d**) During SWS, the extent to which odors evoked category-specific memory reactivation in vmPFC (**a**) and posterior fusiform cortex (**d**) was significantly correlated with behavioral enhancement of cued visuospatial memory retrieval at posttest. fMRI activity is shown at p < 0.001 uncorrected, and images are overlaid on a canonical single-subject T1-weighted MRI scan. (**b,e**) Correlations between fMRI memory reactivation of an individual object category and recall accuracy for that category at posttest, for the two reactivated object categories taken separately in vmPFC (**b**) and posterior fusiform cortex (**e**). Blue and red dots represent individual object categories, where category assignment is arbitrary. (**c,f**) Illustration of correlation depicted in (**a,d**) across time points in vmPFC (**c**) and posterior fusiform cortex (**f**). Time 0 is aligned to odor onset. Error bars depict 90% confidence intervals.

DOI: https://doi.org/10.7554/eLife.39681.014

The following source data and figure supplements are available for figure 3:

**Source data 1.** Relates to panel (b).
DOI: https://doi.org/10.7554/eLife.39681.019
**Source data 2.** Relates to panel (c).
DOI: https://doi.org/10.7554/eLife.39681.020
**Source data 3.** Relates to panel (e).
DOI: https://doi.org/10.7554/eLife.39681.021
**Source data 4.** Relates to panel (f).
DOI: https://doi.org/10.7554/eLife.39681.022
**Figure supplement 1.** Relationship between within-sleep odor-evoked memory reactivation in hippocampus and posttest memory retention.
DOI: https://doi.org/10.7554/eLife.39681.015
**Figure supplement 1—source data 1.** Relates to panel (a).
DOI: https://doi.org/10.7554/eLife.39681.016
**Figure supplement 1—source data 2.** Relates to panel (b).
DOI: https://doi.org/10.7554/eLife.39681.017
**Figure supplement 1—source data 3.** Relates to panel (c).
DOI: https://doi.org/10.7554/eLife.39681.018

([−24,–70, 0], $t_{(16)}$ = 6.73, $p_{FWE}$ = 0.04; *Figure 3d*). When considering each cued category separately, the correlation between reactivation in fusiform cortex and posttest memory performance was evident, but only for one of the odor cues ($r_{1(16)}$ = 0.43, $p_1$ = 0.04; $r_{2(16)}$ = 0.22, $p_2$ = 0.19) (*Figure 3e*). Similar to the time-course profile in vmPFC, within-sleep reactivation in fusiform cortex also emerged following odor onset, but appeared more sustained throughout the duration of odor presentation (*Figure 3f*). Parallel analyses in the hippocampus, which has been previously implicated in memory cueing studies (*Bendor and Wilson, 2012*; *Diekelmann et al., 2011*; *Rasch et al., 2007*;

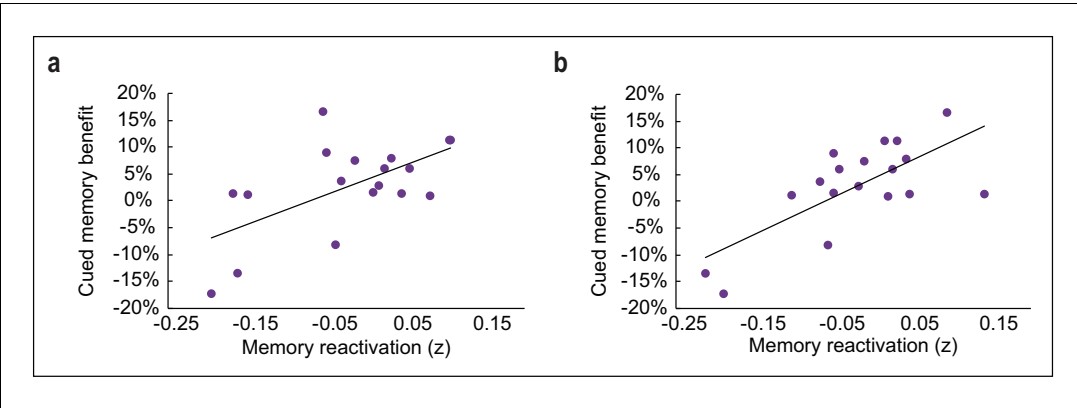

**Figure 4.** Relationship between odor-evoked memory reactivation and cued memory benefit, restricted to cued categories only. (a-b) When limiting the analysis of memory reactivation to a comparison of the two cued object categories (without including the two non-cued object category templates from initial learning for comparison), reactivation measures remained significantly correlated with behavioral memory enhancement in a cluster of interest in vmPFC (a; $r_{(16)}$ = 0.61, p = 0.01) and posterior fusiform cortex (b; $r_{(16)}$ = 0.72, p < 0.001).
DOI: https://doi.org/10.7554/eLife.39681.023

The following source data is available for figure 4:

**Source data 1.** Relates to panel (a).
DOI: https://doi.org/10.7554/eLife.39681.024
**Source data 2.** Relates to panel (b).
DOI: https://doi.org/10.7554/eLife.39681.025

*van Dongen et al., 2012*), did not reveal any association between within-sleep measures of reactivation and subsequent memory performance (*Figure 3—figure supplement 1*).

## Main effect of odors on reactivation of associated category information during sleep

The above analyses were predicated on the idea that among the brain areas that might be reactivated during sleep in response to odor cueing, only those areas that had a systematic effect on post-sleep memory performance would have behavioral relevance. Nonetheless, we also examined whether odor cues evoked reactivation of the associated category per se, without considering memory performance as a covariate. This analysis revealed odor-evoked reactivation of cued category templates in lateral occipital complex ([−56,−62, 4], $t_{(16)}$ = 3.99, $p_{unc}$ = 0.001) (*Figure 5a*) and inferior frontal gyrus ([−50, 30, 8], $t_{(16)}$ = 3.91, $p_{unc}$ = 0.001), although neither of these clusters survived correction for multiple comparisons. Moreover, the degree of memory reactivation in these regions was not correlated with recall accuracy for cued versus noncued objects upon waking in lateral occipital complex ($r_{(16)}$ = −0.33, p = 0.19; *Figure 5b*) or inferior frontal gyrus ($r_{(16)}$ = 0.25, p = 0.32). We also conducted an exploratory analysis to test whether within-sleep reactivation manifests in different brain areas depending on which object category is being cued. By considering each object category separately (rather than collapsing across all four categories), we found that, at a threshold of p < 0.001 uncorrected, odor cues promoted reactivation of associated multivoxel patterns in different brain areas (*Figure 5—figure supplement 1*). However, it is important to note that our experiment was not designed to test individual category effects, because any given category was only cued for 50% of subjects (n = 9). As such, this analysis was underpowered, and therefore these findings should be considered as tentative.

## Within-sleep odor cues activate limbic brain regions

The finding that an index of odor-cued reactivation in vmPFC and fusiform cortex during sleep predicts memory recall necessarily implies that the olfactory system must communicate with extra-olfactory structures to consolidate visuospatial representations. To define potential pathways by which odors can induce cortical reactivation in sleep, we implemented a univariate analysis to characterize

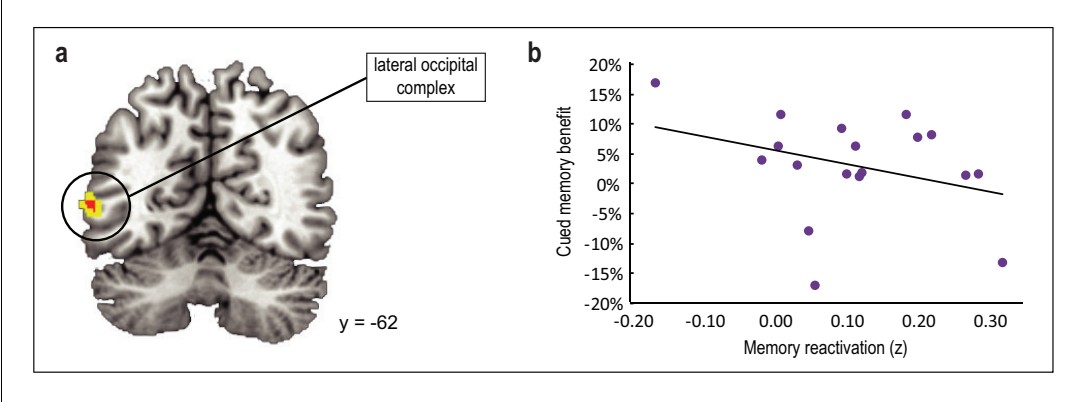

**Figure 5.** Main effect of odor cueing on category reactivation. (a) Odor cues evoked reactivation of associated category templates in lateral occipital complex ([−56,–62, 4], $t_{(16)}$ = 3.99, $p_{unc}$ = 0.001), although this effect did not survive statistical correction for multiple comparisons. (b) The degree of reactivation in this cluster was not predictive of memory enhancement for cued versus noncued categories upon waking ($r_{(16)}$ = −0.33, p = 0.19). fMRI activity is shown at p < 0.001 uncorrected (red) and p < .005 uncorrected (yellow), and images are overlaid on a canonical single-subject T1-weighted MRI scan.

DOI: https://doi.org/10.7554/eLife.39681.026
The following source data and figure supplement are available for figure 5:

**Source data 1.** Relates to panel (b).
DOI: https://doi.org/10.7554/eLife.39681.028
**Figure supplement 1.** Main effect of odor cueing on reactivation of individual object categories.
DOI: https://doi.org/10.7554/eLife.39681.027

which brain areas were activated in the presence of the odor cues, irrespective of their effects on memory performance. We identified an olfactory-related cluster in the left amygdala, extending into left hippocampus ([−28, 0,–24], $t_{(17)}$ = 4.18, $p_{SVC}$ = 0.03), which could plausibly serve as a conduit to cortical structures. Interestingly, this brain area overlaps closely with previous findings showing odor-cued reactivation of left anterior hippocampus during SWS (*Rasch et al., 2007*; *Diekelmann et al., 2011*). A cluster in right amygdala was also observed in an almost symmetrical location to the left amygdala cluster, but did not survive correction for multiple comparisons ([26, 0, -24], $t_{(17)}$ = 3.47, $p_{unc}$ = 0.001).

It is worth pointing out that the identification of olfactory-related regions in sleep could arise because the odor cues were driving downstream activity in the olfactory system, or because they were reactivating previously associated memory content. Ideally, delivery of a control odor that had never been associated with category information would help resolve these two different possibilities. In our study, we opted not to deliver a control odor, to maximize the number of odor trials available for pattern analysis. Therefore, subjects did not receive any odors that were not associated with prior learning during scanning. As such, it was not possible to disentangle contributions of the odor per se, and those related to associated memories. Rather, odor-cued activity in this analysis likely reflects an amalgam of olfactory and reactivation-related influences.

## Odor cues promote connectivity between limbic brain areas and vmPFC during sleep

To assess whether the amygdala-hippocampal cluster might be preferentially coupled with vmPFC or fusiform cortex in the presence of sleep-based odor cues, we used psychophysiological interaction (PPI) analyses to test the functional connectivity between the amygdala-hippocampal cluster and the downstream cortical areas. Significant coupling was observed during odor presentation between this olfactory-related region and vmPFC ($t_{(17)}$ = 1.95, p = 0.03) but was not found between the same region and fusiform cortex ($t_{(17)}$ = 0.85, p = 0.20). The implication is that cued memory reactivation in the sleeping brain begins with odor-evoked activity in medial temporal brain structures, which may mediate the instantiation of visual categorical content in vmPFC. Our data indicate

that an olfactory-prefrontal network may be an important substrate underlying odor-evoked category reactivation.

## Discussion

Here, we used EEG-fMRI recordings combined with multivoxel pattern analysis to investigate the neural mechanisms underlying memory outcomes in a novel olfactory cueing paradigm. First, we demonstrated an olfactory cueing effect, namely, that within-sleep odor cues boost memory performance selectively for associated object categories. Critically, we observed that these behavioral effects are robustly correlated with the degree to which odors promote category-specific reactivation in vmPFC and posterior fusiform cortex, and that these effects hold even when considering both cued object categories separately. Finally, we found that during sleep, odors evoke neural activity in brain areas related to olfactory and limbic function, and that these areas are connected to vmPFC during odor presentation. Together, our findings highlight the functional significance of cue-evoked memory reactivation in promoting consolidation of declarative memories within sleep.

In auditory cueing studies, a multitude of sound cues are routinely presented in sleep to induce memories that are highly specific, and that are often semantically linked to the cue (e.g. cat picture paired with 'meow' sound cue) (*Fuentemilla et al., 2013*; *Oudiette et al., 2013*; *Rudoy et al., 2009*). By contrast, prior olfactory cueing paradigms have largely employed a single arbitrary odor to cue an entire memory task (*Diekelmann et al., 2012*; *Diekelmann et al., 2011*; *Rasch et al., 2007*). Recent studies that utilized two distinct odors during learning and a single olfactory cue during sleep have established that olfactory stimuli can influence behavior with some specificity (*Hauner et al., 2013*; *Rihm et al., 2014*), but to our knowledge ours is the first olfactory study to cue multiple task components during sleep (i.e. two odor cues associated with different object categories). Odors offer unique benefits over sounds as memory cues. Namely, olfactory stimuli are less likely to provoke arousal from sleep (*Carskadon and Herz, 2004*), particularly pure odorants lacking a trigeminal component (*Grupp et al., 2008*; *Stuck et al., 2007*). In addition, the lack of a requisite thalamic relay for odor stimuli and the relative proximity of olfactory and limbic structures in the brain may confer an anatomical advantage (*Gottfried, 2010*). The demonstration that odor cues can promote memory gains more selectively should increase confidence in their utility for more complex memory cueing paradigms.

Perhaps more critically, we show that these selective memory benefits are strongly correlated with the degree to which odors drive reactivation of category information in vmPFC and fusiform cortex in the sleep period. Simultaneous recording of EEG and fMRI data during sleep is technically challenging, and for the vast majority of sensory cueing studies, the sleep period is not scanned. Thus, research exploring fMRI correlates of memory cueing is scant. Two previous EEG-fMRI sleep studies have demonstrated that olfactory cues evoke activity in left anterior hippocampus, at similar coordinates as we observed here (*Diekelmann et al., 2011*; *Rasch et al., 2007*). However, there was no non-cued condition for behavioral comparison in either study, preventing more nuanced conclusions regarding the relationship between brain activity and behavior. One additional study found that auditory cues elicit fMRI activity in parahippocampal cortex during sleep, but did not demonstrate a behavioral effect of cueing on memory performance, perhaps due to subjects' reduced ability to process sound cues in the noisy scanner environment (*Berkers et al., 2018*; *van Dongen et al., 2012*). Although these studies bring important understanding to the dynamics of cueing memories during sleep, our study uniquely highlights its behavioral benefits, permitting us to relate memory performance to neural reactivation on a subject-by-subject basis. Moreover, by utilizing ensemble pattern-based analysis of fMRI data, we were able to probe the contents of within-sleep memory reactivation with greater specificity than would be possible with more conventional fMRI analyses that have been employed in previous studies. In addition, by measuring re-emergence of category-specific brain activity using fMRI, we could pinpoint brain regions participating in neural reactivation with a level of regional and network specificity that EEG approaches cannot provide.

It is worth considering what type of information is being reactivated during sleep. One possibility is that odor cueing leads to reactivation of higher-order representations of object categories that have been established through prior experience with animals, buildings, faces, and tools. An intriguing implication is that if odor cueing in sleep elicits reactivation at the categorical level, then the effects of odor cueing should enhance recall not only for the original group of objects, but also for

any other objects in that same category, and possibly even for related semantic categorical content from any sensory modality. Alternatively, category-specific odors may evoke neural reactivation more specifically, by cueing a subset of objects belonging to that category. Our experimental design does not allow for us to disentangle these two possibilities, but future research should define memory templates during learning at both stimulus and category-specific levels prior to within-sleep cueing to address this question.

Outside of the context of sensory cueing, consolidation is thought to involve the gradual integration of declarative memory traces within the neocortex, guided by the hippocampus. It has been proposed that the PFC may play a key role in integrating memories across cortical modules (*Frankland and Bontempi, 2005*), and indeed frontal lobe damage has been shown to impair recollection (*Simons and Spiers, 2003*). Additionally, medial PFC (mPFC) has been increasingly implicated in remote memory retrieval (*Bonnici and Maguire, 2018*; *Euston et al., 2012*; *Jin and Maren, 2015*; *Preston and Eichenbaum, 2013*; *Takashima et al., 2007*), especially when memory encoding is followed by a period of sleep (*Gais et al., 2007*; *Sterpenich et al., 2009*). It has also been suggested that sleep-based integration of memories into neocortical networks is aided by the slow oscillations characteristic of SWS (*Rasch and Born, 2013*). In line with this concept, mPFC is thought to be a predominant generator of slow wave activity (SWA) (*Murphy et al., 2009*), and cortical volume loss in mPFC has been linked to parallel deficits in SWA and sleep-dependent memory retention (*Mander et al., 2015*; *Mander et al., 2013*). Our findings that odors presented during SWS may drive reactivation of associated mnemonic content in vmPFC to support recall meshes well with this prior work, providing robust mechanistic support that cue-evoked cortical reactivation promotes within-sleep consolidation in the human brain.

## Materials and methods

### Subjects

Thirty-two healthy human subjects (21 female; *mean age* = 25.25 years; *age range* = 19–37 years) gave written informed consent to take part in the study, which was approved by the Institutional Review Board at Northwestern University. All subjects were right-handed non-smokers under 40 years of age who had undergone fMRI scanning at least once prior to the study, and reported that they thought they could feel relaxed and fall asleep during fMRI scanning. Additional criteria for exclusion included history of significant medical or psychiatric illness, history of sleep disorder, use of psychotropic medications, nasal congestion, and frequent snoring. Subjects were required to go to bed at their habitual bedtime the night before the main experiment, and to wake up three hours earlier than their habitual wake time the following morning. Sleep-wake activity was monitored via actigraphy (Spectrum, Phillips) during the night prior to the main experiment, and subjects completed an online sleep diary (adapted from the National Heart, Lung, and Blood Institute) for 1 week prior to the main experiment. Subjects were also asked to refrain from napping, and from consuming caffeine or alcohol on the day of the main experiment.

Eleven subjects were excluded from analysis due to insufficient or fragmented SWS during the nap phase, which prevented the experimenters from presenting each of the two odor cues a minimum of 14 times. An additional three subjects were excluded due to arousal and subsequent odor perception during the nap phase. Eighteen subjects, equal to 56.25% of all subjects, were retained for analysis (11 female; *mean age* = 25.11 years; *age range* = 19-37), in line with prior studies reporting subjects' ability to fall asleep and stay asleep in the fMRI scanner environment (*Diekelmann et al., 2011*; *van Dongen et al., 2012*).

### Stimuli

Visual stimuli consisted of 128 high-resolution portable network graphic (PNG) images obtained from the internet. Images were cropped and displayed on a grey background with identifying labels. Images included 32 well-known exemplars from each of four categories: animals (e.g. camel, zebra), buildings (e.g. Eiffel Tower, Taj Mahal), faces (e.g. Barack Obama, Emma Watson), and tools (e.g. screws, wrench). To ensure that exemplars were familiar to subjects, an independent group of 12 subjects provided labels for each image of a larger stimulus set, and the most easily identifiable images were retained for use in the study. Thirty-two additional scrambled images were generated

from a subset of category images, and accompanying labels consisted of arbitrary combinations of letters. Scrambled images were included as a localizer, to allow for identification of functionally defined voxels if needed.

Olfactory stimuli comprised four easily distinguishable, familiar odorants. To ensure that odor cues could be easily discriminated from each other, four odorants were selected from a larger set of eight well-known odorants (banana, cedar, cinnamon, garlic, lemon, mint, rose, vanilla) on an individual subject basis (see next section). Odorants were delivered by a custom 12-channel computer-controlled olfactometer at a flow rate of 6.72 L/min via a Teflon tube secured beneath the nose.

## Odor selection

The day before the main experiment, each subject took part in an odor selection task so that we could determine which four odors were most discriminable. Immediately prior to the task, subjects were familiarized with eight odors and their respective labels. Subjects then made pairwise similarity ratings between all possible pairs (28 unique pairs). Each odor pair was presented two times, for a total of 56 trials. During each trial, subjects were cued to sniff two consecutive odors presented 4.5 s apart and then to make a pairwise similarity rating on a visual analog scale from 'extremely different' to 'extremely similar'. A 'pairwise similarity score' was calculated for each odor pair by taking the average similarity rating across two trials. Based on pairwise similarity scores, four odors were selected to minimize perceptual overlap (i.e., minimal pairwise similarity scores for odor pairs included in the final set of four). Thus, a 'total similarity score' was computed for each possible combination of four odors (70 total combinations). For instance, for a set of four odors (A, B, C, and D), with all possible pairwise combinations among these odors, the total similarity score would be the sum of similarity ratings for A vs. B, A vs. C, A vs. D, B vs. C, B vs. D, and C vs. D. The set of four odors with the lowest total similarity score was retained for the main experiment.

## Main experiment

The main experiment lasted approximately 6.5 hr and took place at night, such that the nap phase aligned with each subject's usual bedtime, to increase the likelihood of subjects falling asleep during fMRI scanning. The main experiment began between 7 pm and 10:30 pm, and ended between 1:30 am and 5 am.

## Odor ratings

Upon arriving at the MRI facility, subjects rated the four selected odors in terms of intensity (from 'barely detectable' to 'extremely strong') and valence (from 'extremely unpleasant' to 'extremely pleasant'). Odors selected as within-sleep cues versus those not selected did not differ in perceived intensity ($t_{(17)}$ = 0.60, p = 0.56) or valence ($t_{(17)}$ = 0.18, p = 0.86).

## Initial learning

Subjects were instructed to learn the locations of visual stimuli (category images and scrambled images) on a grey 4-x-4 grid while undergoing fMRI scanning. Each of the images appeared a total of three times over the course of the task, which was divided into 12 2.25 min runs. Each run consisted of five blocks (animals, buildings, faces, tools, scrambled) presented in a random order, with 12 s between blocks. Each block lasted 14 s, and consisted of a series of eight images presented on the grid for 1 s each, with 0.75 s between consecutive image presentations. Grid locations were balanced such that two objects per category were presented in each of the 16-grid spaces.

## Odor-category association

Subjects were then led from the scanner to a testing room, where they learned to associate each of the four odors with each of the four object categories (e.g. rose odor + building images). Odor-category pairs were randomly assigned for each subject. On each trial, subjects were cued to sniff upon odor presentation, and then immediately afterwards one object from each of the four categories appeared in the four different quadrants of the screen. Category objects were identical to those presented during initial learning. Subjects were instructed to select the object from the associated category as quickly and accurately as possible, and then a green box appeared around the correct choice for 2 s, as feedback. Trials were spaced at least 6 s apart, to avoid habituation and odor

cross-contamination. The task continued until each odor-category pair (e.g. rose odor + building images) was correctly identified 16 times (number of trials for perfect performance = 64), to ensure robust odor-category associations. Subjects learned these associations rapidly (*mean* = 69.44, ±0.74 SEM, *range* = 66–78 trials to reach criterion).

### Learning to criterion

Subjects then continued to learn the same object locations as in the initial learning phase, with two key differences: (1) objects appeared in the presence of category-specific odors, and (2) subjects were actively tested on their knowledge of object locations. During each trial, an object appeared in the center of the screen for 0.5 s, and then subjects attempted to select the grid space where the object belonged (16 grid spaces, chance = 6.25%). Then, the object appeared in the correct grid space for 0.5 s, as feedback. Objects were presented in category blocks of eight objects per category during continuous presentation of the associated odor. Blocked presentation of objects allowed for efficient delivery of associated odors, as blocks were spaced 12 s apart to avoid habituation and odor cross-contamination. The task continued until subjects placed each object in the correct grid space once (*mean* = 250.22, ±12.64 SEM, *range* = 178–367 trials to reach criterion). To ensure continued attention to both odor and object category stimuli and associations, subjects performed 16 'catch' trials (four per odor-category pair) over the course of the task. During catch trials, subjects selected the category that belonged with the presented odor, and then received feedback (identical to the odor-category association task above). Subjects retained a strong knowledge of odor-category associations during catch trials (*mean* = 94.79% correct, ±1.69% SEM).

### EEG cap

Subjects were fitted with an MRI-compatible EEG cap (BrainCap MR, Brain Products; see sleep recording section for more details).

### Pretest

While wearing the EEG cap and immediately prior to the nighttime nap, subjects were tested outside the fMRI scanner on their knowledge of object locations. During this phase, each object appeared in the center of the screen for 0.5 s, and subjects selected the grid space where they believed the object belonged, without receiving feedback. As in the previous phase, objects were presented in category blocks of eight objects per category during continuous presentation of the associated odor, and blocks were spaced 12 s apart. Again, subjects performed 16 catch trials (four per odor-category pair) over the course of this pretest session. Catch trials were identical to those of the previous task, except that subjects did not receive feedback. During catch trials, subjects continued to demonstrate excellent retention of odor-category associations (*mean* = 95.83%, ±1.13% SEM).

### Sleep

Subjects were instructed to relax and try to fall asleep during continuous fMRI scanning. At the start of the scan, subjects were given the option to take part in a monotonous reaction time task for approximately 5 min, to help them re-acclimate to the scanning environment. Subjects were instructed to press an MRI-compatible button each time a central crosshair changed color, but to disengage from the task if they felt very drowsy. Subjects were further instructed to press the button during the nap if they perceived an odor. During stages 2 and 3, two of the four task-related odors were presented in alternating 16 s on/16 s off blocks (to prevent habituation), as category-specific cues (*mean* = 50.61, ±4.61 SEM, r*ange* = 30–105 total odor presentations). To decrease the chances of subjects waking during odor presentation, experimenters waited to observe a minimum of 2 min of continuous stage 2 sleep prior to initiating odor delivery. Odors were selected as within-sleep stimuli strategically, to ensure that cueing of categories was counterbalanced across subjects (six possible category pairs, each presented to three of the 18 subjects). After the nap, subjects exited the scanner and took a quick shower, to rinse EEG gel from their hair and to overcome sleep inertia. For the remainder of the main experiment, subjects were not exposed to odorants.

### Interference learning and test

Approximately 30 min after waking, subjects learned new grid locations for the same objects as presented in initial learning. Task structure was identical to that of initial learning, except that intervals between category blocks were limited to 4 s, since longer intervals were unnecessary in the absence of odors. Subjects were allowed a 5 min break after interference learning, and then they were tested on their knowledge of the new object locations (without feedback). The interference test was identical to the pretest, except that intervals between category blocks were limited to 4 s, and there were no odors or catch trials.

### Posttest

After a 30-min break, subjects completed a posttest to assess their knowledge of the original (non-interference) object locations. Task structure was identical to that of the interference test.

### One-week follow-up

One week after the main experiment, subjects returned for a series of follow-up memory tests. Although the follow-up visit was scheduled in advanced, subjects were not given information regarding the nature of follow-up tasks. First, subjects completed a free recall test, in which they were given two minutes per category to list as many exemplars from that category as they could remember. Next, subjects were tested on their retention of the original object locations in a task that was identical to the posttest from the main experiment. There was no difference between cued and non-cued categories for free recall ($t_{(17)}$ = 0.20, p = 0.42), change in spatial recall from pretest baseline ($t_{(17)}$ = 0.17, p = 0.43), or RTs ($t_{(17)}$ = 0.46, p = 0.65) during follow-up memory tests. Finally, subjects were verbally instructed to sniff each odorant, and to recount which category it was paired with during the main experiment. Seventeen of 18 subjects could correctly recall odor-category associations.

### Sleep recording

During the sleep session, EEG data were collected with an MRI-compatible EEG system (BrainAmp MR Plus, Brain Products) in order to restrict odor delivery to stages 2 and 3 of sleep. The 32-channel EEG cap contained 26 scalp electrodes and two electrooculography (EOG) electrodes, and wires to connect electrodes for chin electromyography (EMG) and electrocardiography (ECG). EEG data were sampled at 5 kHz, scanner and cardioballistic artifacts were removed online using Brain Products software (Rec View), and data were scored in accordance with standard criteria (*Silber et al., 2007*) (*Table 1*). To compare power spectral density of EEG data across conditions (i.e., odor-on versus odor-off; stage 3 sleep versus stage 1 sleep), we employed a fast Fourier transform analysis of frontal electrode 'Fpz' using the 'pwelch' function in Matlab.

### fMRI data acquisition

MRI data were collected with a 3-Tesla scanner (Siemens PRISMA) equipped with a 64-channel head coil, using T2-weighted echoplanar imaging. Each volume comprised 40 slices covering the whole brain (field of view, 210-x-203 mm; matrix size, 124-x-120 voxels; slice thickness, 3 mm; in-plane resolution, 1.69-x-1.69 mm; repetition time, 2500 ms; echo time, 25 ms; flip angle, 80°). An additional whole-brain anatomical T1-weighted MRI scan was acquired for coregistration purposes (GRAPPA; voxel size, 0.8 mm$^3$).

### fMRI preprocessing

MRI data were preprocessed and analyzed using SPM12 (http://www.fil.ion.ucl.ac.uk/spm/). Functional images were realigned to the mean of the images, motion corrected, and coregistered to the T1-weighted image. For univariate and PPI analyses, images were normalized and then spatially smoothed with a 6 mm Gaussian kernel. Multivariate searchlight analyses were conducted in each subject's native space, and images were minimally smoothed with a 2 mm Gaussian kernel. Searchlight analyses were restricted to grey matter voxels, by generating a grey matter mask from the SPM12 tissue probability map, and then warping that mask to the subject's native space using the transformation parameters from the standard T1 template to the subject's individual T1-weighted image.

## Multivoxel pattern analysis: selection of category-sensitive voxels

To identify category-sensitive voxels, a GLM was constructed for each subject from initial learning scans, where category images were modeled as five separate regressors of interest (animals, buildings, faces, tools, scrambled) in a blocked design. Nuisance regressors included the six motion parameters generated from realignment, and beta estimates were calculated for each condition. To quantify pattern discrimination of the four categories during initial learning, a whole-brain searchlight-based correlation analysis was implemented. At each search sphere (radius = 3.7 voxels), 48 beta pattern vectors were extracted (four object categories X 12 runs). In a leave-one-out approach, beta patterns from 11 'training' runs were averaged, and compared to beta patterns from the remaining 'test' run. In each iteration, the mean across conditions was first subtracted from training and test beta patterns separately. This was followed by calculation of the linear correlations between all training and test patterns, and then the resulting correlation coefficients were Fisher's $Z$ transformed. This procedure was repeated a total of 12 times, so that each run could be left out in turn as a test run. The resulting values were averaged across iterations. As an index of pattern discrimination, within-category correlations (*training* patterns versus the category-congruent *test* pattern) were compared to the average of between-category correlations (training patterns versus the three category-incongruent test patterns). This procedure was repeated at each search sphere, and the resulting pattern discrimination maps were normalized and then smoothed with a 6-mm Gaussian kernel prior to group comparison. Category-sensitive voxels were designated at the group-level, from the contrast within-category correlations > between-category correlations (p < 0.001). Critically, the next analysis was restricted to brain regions that demonstrated category-selectivity during learning (see below), as this was a prerequisite to identifying reactivation of the same category information during sleep.

In a follow-up analysis to determine the selectivity for each object category separately, the same steps were repeated with one exception. Rather than collapsing indices of pattern discrimination across categories, within-category correlations were compared to the average of between-category correlations for each object category separately. This resulted in four group-level pattern selectivity maps, one per object category.

## Multivoxel pattern analysis: re-emergence of category information during sleep

To determine whether category information re-emerged in response to odor cues during sleep, the same GLM that was used for selection of category-specific voxels (see previous section) was used to construct templates of the four object categories. An additional sleep-based GLM was constructed for each subject, where each of the two odor cues were modeled as separate regressors of interest. Odor onset times were adjusted to align with the first point of inhalation (i.e. rising slope on the breathing trace) after odor presentation. To minimize signal contributions induced by head motion during the sleep scan, all volumes prior to the first odor presentation and following the last odor presentation were discarded (with the exception of six additional volumes on either end). Given the large number of volumes remaining (*mean* = 948.61, *range* = 401–1416), there were sufficient degrees of freedom in our GLM to incorporate extra nuisance regressors without overfitting. This allowed us to account for head motion more rigorously given the long duration of the sleep session. (In contrast, this more rigorous motion correction was not possible for the wake sessions without overfitting, due to the smaller number of fMRI volumes (58 per session), and thus fewer degrees of freedom.) Nuisance regressors included the six motion parameters generated from realignment, and their squares, derivatives, and squared derivatives (24 total). To account for within-scan motion, the signal difference between even and odd slices, the within-volume variance across slices, and derivatives of both parameters were also included as nuisance regressors. Additional nuisance regressors were included when necessary, to capture individual volumes demonstrating excessive head motion. Finally, the respiration trace was post-processed and down sampled to the scanner repetition time frequency to be included as a nuisance regressor. Beta estimates were calculated for each of the two odor conditions.

To search for pattern-based reactivation during sleep, a whole-brain searchlight-based correlation analysis was implemented (as in voxel selection). At each searchlight sphere, template images were constructed from the initial learning data for each of the four categories by averaging the extracted

beta pattern vectors across runs for each condition (category) separately, and then subtracting the mean across conditions from each voxel. Next, two beta pattern vectors were extracted from the sleep period, each corresponding to one of the two odor cues (mean activity was not subtracted from each voxel here, since doing so across the two cues would have created two anticorrelated vectors, essentially reducing two data points to a single data point). Correlation coefficients were calculated between those odor-cued activity pattern vectors and each of the four category templates separately, and the resulting *r* values were converted to Fisher's *Z* scores. As an index of memory reactivation, the cued correlation (sleep activity pattern versus cued learning template) was compared to the average of non-cued correlations (sleep activity pattern versus the three non-cued learning templates) for each beta pattern vector separately, and then the resulting values were averaged across the two conditions. This procedure was repeated at each search sphere, and the resulting memory reactivation maps were normalized and then smoothed with a 6-mm Gaussian kernel prior to group comparison. A group-level correlation analysis was conducted, in which cued memory benefit was included as a covariate of interest, and category-selective voxels (identified during prior learning) were designated as an explicit mask. Cued memory benefit was defined as visuospatial memory performance at posttest (expressed as a percentage of items remembered from pretest baseline) for cued minus non-cued categories.

In a follow-up analysis, the same steps were repeated for the two reactivated categories separately (without considering the two non-reactivated categories). Memory reactivation maps were constructed, and values were extracted from clusters of interest in vmPFC and fusiform cortex (identified from the multivoxel correlation analysis, p < 0.001), and then averaged. The resulting values were correlated with posttest recall accuracy (again, expressed as a percentage of items remembered from pretest baseline) for the two reactivated categories individually.

To reveal the time course of reactivation, a separate sleep-based finite impulse response (FIR) model was constructed. Ten regressors of interest, spaced apart at 2.5 s intervals, were included for each of the two odorants, spanning a 25 s time window. The regressors modeled responses starting three volumes prior to odor onset, a single volume aligned to odor onset, and six volumes following odor onset. The model also included the same nuisance regressors described above. For regressors corresponding to each time point, beta estimates were extracted for each of the two odors. Next, a searchlight analysis was conducted (as in the main analysis), resulting in ten separate memory reactivation maps (one per time point). Again, values were extracted from the resulting reactivation maps for both vmPFC and fusiform cortex clusters, and then correlated with cued memory benefit at each respective time point.

Contrary to prediction, searchlight analyses did not reveal a relationship between memory reactivation in the hippocampus and memory performance. To further investigate effects in this brain area, all three analyses outlined above were repeated for an ROI in bilateral hippocampus taken from the AAL Atlas (as an alternative to the whole-brain searchlight approach).

Finally, two additional analyses were conducted to elucidate the main effect of within-sleep odors on reactivation of associated object categories, without considering cued memory benefit as a covariate. The first analysis was derived from the same model that included cued memory benefit as a covariate of interest, but here the main effect was considered instead of its relationship with memory performance. Again, results were masked according to category-selective voxels identified from the initial learning phase. A second analysis was conducted to determine whether there were spatial differences in reactivation maps for the four object categories. Here, instead of considering all four object categories across all 18 subjects, four separate second-level models were constructed, and each included reactivation maps for a single object category. Because each of the 18 subjects received two odor cues during sleep, this meant that each category was only cued for nine subjects. Thus, a total of nine reactivation maps were included in each of the four second-level models. Given the limited number of subjects included in each model, the resulting category maps were not constrained by a mask.

## Univariate analysis: odor-evoked activity in sleep

In an additional analysis, we aimed to identify brain regions activated by odor cues during sleep. The GLM used to identify voxels activated by odor cues was identical to the GLM constructed from the sleep period for the multivoxel analysis described previously, except that the model was applied to images that had been normalized and smoothed with a 6 mm Gaussian kernel. Individual contrast

maps for odor onset were computed at the subject level and tested at the group level using a one-sample $t$ test.

## PPI analysis: connectivity during odor presentation

The gPPI toolbox (*McLaren et al., 2012*) was employed to measure connectivity between hippocampus and vmPFC during odor presentation. We estimated a PPI model for each subject using normalized, smoothed images. The physiological factor was defined as fMRI activity in a seed region (5-mm sphere surrounding the peak voxel in amygdala/hippocampus identified from the univariate analysis), and odor onset was the psychological factor. Voxel-wise connectivity parameters at odor onset were extracted from the same vmPFC and posterior fusiform clusters as previously and averaged across voxels for each subject individually, and then a one-sample $t$ test was implemented for group-level comparison.

## Statistical analysis

To test our a priori prediction that memory retention for cued information would be enhanced compared to non-cued information, as in previous cueing studies (*Oudiette and Paller, 2013*; *Rasch et al., 2007*; *Rudoy et al., 2009*; *Schouten et al., 2017*; *Shanahan and Gottfried, 2017*; *Spiers and Bendor, 2014*), we used a one-tailed Wilcoxon signed-rank test and a binomial test. To analyze RTs from the interference test, trial-by-trial RTs were log transformed, and compared for cued versus non-cued category objects across subjects using a two-tailed paired $t$ test. To determine the effects of odor cues (odor-on versus odor-off) and sleep stage (stage 3 versus stage 1) on EEG power spectral density, we performed two-factor (condition-x-frequency) repeated-measures ANOVAs (two-tailed). To identify brain regions containing category information, and where reactivation of category information was correlated with cued memory benefit, we conducted one-tailed $t$ tests on normalized and smoothed searchlight maps. p-Values were defined at the peak voxel, and adjusted for the family-wise error rate based on the explicit initial learning mask (i.e. voxels that contained category information at initial learning). In follow-up tests, reactivation values were extracted from searchlight maps for each subject, and one-tailed linear correlations with behavioral recall measures were computed. In the univariate analysis used to identify voxels activated during odor presentation, we again used a one-tailed paired $t$ test of normalized and smoothed contrast maps, and p-values were defined at the peak voxel, corrected for the family-wise error rate based on an olfactory region of interest in left amygdala from the AAL Atlas. Finally, for the PPI analysis, one-tailed $t$ tests were used to compare connectivity values extracted from a cluster of interest in vmPFC to zero.

## Acknowledgements

We thank J Howard and X Bao for helpful guidance in analysis, and T Parrish for technical support.

## Additional information

### Funding

| Funder | Grant reference number | Author |
| --- | --- | --- |
| National Institute on Deafness and Other Communication Disorders | F31DC015374 | Laura K Shanahan |
| National Institute of Neurological Disorders and Stroke | T32NS047987 | Laura K Shanahan |
| Brain and Behavior Research Foundation | 23272 | Jay Gottfried |
| National Institute on Deafness and Other Communication Disorders | R01DC010014 | Jay Gottfried |

The funders had no role in study design, data collection and interpretation, or the decision to submit the work for publication.

### Author contributions
Laura K Shanahan, Conceptualization, Data curation, Software, Formal analysis, Funding acquisition, Validation, Investigation, Visualization, Methodology, Writing—original draft, Writing—review and editing; Eva Gjorgieva, Investigation: Critically and substantially involved in nighttime data collection, which involved piloting, optimizing, and troubleshooting setup and implementation of the EEG-fMRI equipment; Ken A Paller, Conceptualization, Writing—review and editing; Thorsten Kahnt, Supervision, Visualization, Writing—review and editing; Jay A Gottfried, Conceptualization, Resources, Supervision, Funding acquisition, Visualization, Writing—original draft, Project administration, Writing—review and editing

### Author ORCIDs
Laura K Shanahan (iD) http://orcid.org/0000-0001-6050-425X
Thorsten Kahnt (iD) http://orcid.org/0000-0002-3575-2670

### Ethics
Human subjects: Human subjects gave informed consent to take part in the research study, which was approved by the Institutional Review Board at Northwestern University (study ID: STU00094974).

### Decision letter and Author response
Decision letter https://doi.org/10.7554/eLife.39681.035
Author response https://doi.org/10.7554/eLife.39681.036

## Additional files

### Supplementary files
• Transparent reporting form
DOI: https://doi.org/10.7554/eLife.39681.031

### Data availability
Source data files have been provided for all main text and supplementary figures. fMRI statistical maps for all main text and supplementary figures are available on NeuroVault (https://neurovault.org/collections/DODNGFFT/).

The following dataset was generated:

| Author(s) | Year | Dataset title | Dataset URL | Database and Identifier |
| --- | --- | --- | --- | --- |
| Shanahan LK, Gjorgieva E, Paller KA, Kahnt T, Gottfried JA | 2018 | Odor-evoked category reactivation in human ventromedial prefrontal cortex during sleep promotes memory consolidation | https://neurovault.org/collections/DODNGFFT/ | NeuroVault, 4060 |

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
