## [Decision Letter]

Thank you for submitting your article "Odor-evoked replay in human prefrontal cortex promotes memory consolidation during sleep" for consideration by *eLife*. Your article has been reviewed by two peer reviewers, and the evaluation has been overseen by a Reviewing Editor and Timothy Behrens as the Senior Editor. The following individual involved in review of your submission has agreed to reveal his identity: Björn Rasch (Reviewer #1).

The reviewers have discussed the reviews with one another and the Reviewing Editor has drafted this decision to help you prepare a revised submission.

Summary:

The authors examined the effect of odors on content-specific reactivation patterns during sleep in healthy human volunteers. Participants first learned a visual object-location task involving four categories of different objects. During this training, category-specific brain activation was identified for each participant using fMRI. Participants then associated each of the four categories with four different odors and learned the object-location task to criterion. Participants were re-exposed to two of the four odors during subsequent sleep in the fMRI scanner and recall was tested afterwards. The authors show that during sleep, odor re-exposure activated category specific voxel patterns. Furthermore, the degree of category-specific reactivation during sleep in the medial frontal cortex and the fusiform gyrus predicted post-sleep memory performance. Brain activity in the hippocampus did not show signs of content-specific reactivation.

Essential revisions:

1) The authors exclusively focus their analysis on the predictions of signs of reactivation during sleep on behavioral memory performance, which is of course interesting and relevant. However, it did not become clear to me whether they are actually able to find patterns of reactivation during sleep. The most crucial test for detecting memory reactivation in this paradigm is not reported.

The authors claim to detect reactivation in fMRI and state it allows them to characterize the process more accurately than previous work done e.g. in EEG. They claim they study reactivation as patterns observed during wakefulness reinstated by odor cueing during sleep (which has not/only recently been shown previously in humans). However, at no point in their work do the authors show actual reactivation of learning related activity. They do not report a significantly stronger reinstatement of cue-related category patterns than non-related category patterns after cue presentation during sleep, which is the crucial test in this design. The authors instead relate a non-substantiated indirect measure of memory reactivation (which is arguably an approach to take) to a behavioural measure and perform statistical tests on this second-order measure. I am not against publication of this work, but it must be made clear if the authors cannot detect significant reactivation of category content upon reactivation cues. They should report the straightforward analysis of testing within-category similarity vs. between-category similarity for learning activity and post memory cue activity during sleep and make it clear should this direct test for cue-related memory reactivation fail (showing no significant difference). I strongly believe the rest of the analysis to be worthwhile and informative in its own way, yet this issue needs to be made transparent and the very strong claims need to be toned down accordingly (relating content-specific brain activity between wake and sleep, detecting "replay/reactivation").

2) The authors should report the general results of the identification of category-specific voxels during sleep in more detail, i.e. the main effect of the memory replay maps. In which area did they find the highest difference in correlation measures? Did they observe a difference in spatial distribution in the replay maps when the two odor cues were considered separately?

3) Please report also more details on the general results of the category-specific voxels during training. What are the brain areas with the highest category specificity? Did one of the categories contribute more to this separation than other categories? Was the categorization for some categories different in specific brain areas?

4) The methodological approach appears to suitable to use a classifier approach on the data. Could the authors train a classifier on their training data and observe above chance classification of cued categories during sleep? Otherwise, please discuss reasons why the authors did not choose to do so and explain the advantages of the approach reported in the manuscript.

5) The authors state they study "memory replay". Replay is commonly used in animal literature to refer to sequential reactivation of firing patterns (as observed during learning) and many groups are currently striving to detect something similar in humans. The current work does not show sequential replay, it only shows that an indirect measure of memory reactivation is related to memory performance. They should thus refrain from using the term replay.

6) Why was the rigorous way to ensure no artefacts by head motion applied to sleep data not similarly applied to wake data where more head movements can be expected? Please justify.

7) Why does the sleep analysis not follow the same rationale as the wake analysis, first subtracting a mean of both "categories" (here cue responses), then calculating pairwise correlations?

8) Figure 2B. The behavioural reaction time measure might be confounded by accuracy. Were there differences in number of correct trials for interference learning for cued vs. non-cued items? Do reaction time results hold if only correct trials are analysed? If that difference exists and the reaction time results do not hold when only correct trials are analysed, please report this in the paper with a note of caution.

---

## [Author Response]

Essential revisions:1) The authors exclusively focus their analysis on the predictions of signs of reactivation during sleep on behavioral memory performance, which is of course interesting and relevant. However, it did not become clear to me whether they are actually able to find patterns of reactivation during sleep. The most crucial test for detecting memory reactivation in this paradigm is not reported.The authors claim to detect reactivation in fMRI and state it allows them to characterize the process more accurately than previous work done e.g. in EEG. They claim they study reactivation as patterns observed during wakefulness reinstated by odor cueing during sleep (which has not/only recently been shown previously in humans). However, at no point in their work do the authors show actual reactivation of learning related activity. They do not report a significantly stronger reinstatement of cue-related category patterns than non-related category patterns after cue presentation during sleep, which is the crucial test in this design. The authors instead relate a non-substantiated indirect measure of memory reactivation (which is arguably an approach to take) to a behavioural measure and perform statistical tests on this second-order measure. I am not against publication of this work, but it must be made clear if the authors cannot detect significant reactivation of category content upon reactivation cues. They should report the straightforward analysis of testing within-category similarity vs. between-category similarity for learning activity and post memory cue activity during sleep and make it clear should this direct test for cue-related memory reactivation fail (showing no significant difference). I strongly believe the rest of the analysis to be worthwhile and informative in its own way, yet this issue needs to be made transparent and the very strong claims need to be toned down accordingly (relating content-specific brain activity between wake and sleep, detecting "replay/reactivation").

We would like to thank the reviewers for their positive comment that our analysis of the relationship between signs of reactivation and behavioral memory performance is interesting and relevant. Our central analysis was indeed focused on behavioral correlates, as we reasoned that signs of neural reactivation should be strongest in those subjects whose memories were most susceptible to sensory cueing.

That said, we agree that the ability of cues to evoke pattern reinstatement per se (without considering behavior as a covariate) is also a crucial test in our design. Interestingly, this analysis revealed a main effect of cueing on pattern reactivation in the lateral occipital complex and the inferior frontal gyrus. These clusters were observed at a threshold of *p* < 0.001, though did not survive correction for multiple comparisons across category-selective voxels identified during the initial learning phase. These new analyses and findings, along with a new Figure 5, can be found in the subsection “Main effect of odors on reactivation of associated category information during sleep” and the subsection “Multivoxel pattern analysis: re-emergence of category information during sleep”. We also uploaded the brain map from this analysis to our Neurovault repository so it is publicly available. Moreover, where appropriate, we have toned down our claims regarding reactivation per se versus reactivation related to behavioral memory improvements.

2) The authors should report the general results of the identification of category-specific voxels during sleep in more detail, i.e. the main effect of the memory replay maps. In which area did they find the highest difference in correlation measures? Did they observe a difference in spatial distribution in the replay maps when the two odor cues were considered separately?

This comment is highly relevant to comment #1 above. To reiterate, we have conducted analyses to test the main effect of odor cueing on category reactivation.

The reviewers also ask an intriguing question as to whether the two odor cues lead to reactivation in different brain areas. To clarify, each subject received two odor cues (out of a total of four possible cues) during sleep to reactivate information from two of the four object categories. Cues were counterbalanced, to ensure that each object category was represented equally (nine times total) across subjects. In this manner, the first and second odor cues are arbitrary, in that they were not consistently paired with the same two object categories for every subject. Thus, directly comparing effects evoked by the two odors doesn’t lend itself to a clear interpretation.

That said, it is possible to consider the reactivation maps for each of the four object categories separately, and doing so could be informative if reactivation manifests in different brain areas depending on the cued object category. To test this question, we constructed four separate group-level models, each incorporating reactivation maps for a single object category. Interestingly, odors appeared to promote pattern reactivation in a unique brain area for each of the object categories. For instance, at a threshold of *p* < 0.001 uncorrected, animal and tool patterns were reinstated in different parts of parahippocampal cortex, building patterns were reinstated in lateral occipital complex, and face patterns reemerged in occipital cortex. We describe this new analysis in the revised manuscript (subsection “Main effect of odors on reactivation of associated category information during sleep” and “Univariate analysis: odor-evoked activity in sleep”), and summarized preliminary findings in Figure 5—figure supplement 1. We also uploaded brain maps from this analysis to our Neurovault repository.

While these new findings bring additional insights to category-specific regional contributions of pattern reactivation, it is important to note that our experiment was not designed to test individual category effects, because any given category was only cued for 50% of subjects (n = 9). As such, this analysis was underpowered, and therefore these findings should be considered as tentative.

3) Please report also more details on the general results of the category-specific voxels during training. What are the brain areas with the highest category specificity? Did one of the categories contribute more to this separation than other categories? Was the categorization for some categories different in specific brain areas?

We thank the reviewers for this opportunity to elaborate on our results regarding category specificity during the initial learning phase of the experiment. To provide a more transparent understanding of which brain areas were most robustly associated with category-specific activity, we replaced the binary mask image from Figure 1C with a heat map, and have also provided a new supplementary figure (Figure 1—figure supplement 1). To disentangle the contributions of individual object categories, we conducted a voxel-wise pattern analysis where category-selective maps were computed for each of the four object categories separately (rather than averaged across categories, as in the original analysis). This approach revealed category-specific encoding across widespread brain areas for all four categories. We summarized these findings in the new Figure 1—figure supplement 1, and incorporated methods details in the revised text (subsection “Multivoxel pattern analysis: selection of category-sensitive voxels”). Individual selectivity maps for each object category are available on Neurovault.

4) The methodological approach appears to suitable to use a classifier approach on the data. Could the authors train a classifier on their training data and observe above chance classification of cued categories during sleep? Otherwise, please discuss reasons why the authors did not choose to do so and explain the advantages of the approach reported in the manuscript.

The reviewers bring up a pertinent question about an alternative classifier approach. We did consider employing such an approach in our analysis, but ultimately decided that a correlation-based method would be more appropriate here. Correlation-based similarity measures fall on a continuous spectrum, whereas classifiers yield a binary output (“correct” or “incorrect”). Because we were interested in comparing a continuous behavioral metric (recall accuracy) to pattern reinstatement, we felt strongly that correlation-based pattern analysis was the more fitting tool.

However, we do acknowledge that a classifier approach could be applied here. To fully respond to this comment, we re-ran our main analysis, this time comparing recall behavior to classifier accuracy. We trained a four-way support vector machine on the four object categories from the initial learning phase, and then tested it on the sleep data. Note that because the sleep data only contained two data points for each subject (one parameter estimate per within-sleep odor cue), the classifier could effectively be tested on only two items. As such, computing an average classification accuracy across two data points could only result in one of three possible outputs (0, 0.5, or 1), providing poor resolution for comparison to the continuous behavioral variable. Nonetheless, we still observed a correlation between classifier accuracy and recall in both ventromedial prefrontal cortex (vmPFC) and posterior fusiform cortex at very similar coordinates as found in our original correlation-based analysis (vmPFC: [-6, 50, -12], *t*_(16)_ = 3.87, *p*_unc_ = 0.001; posterior fusiform cortex: [-28, -74, 6], *t*_(16)_ = 4.72, *p*_unc_ < 0.001), though effects were weaker, possibly due to the reduced resolution of the SVM classifier.

Given our rationale for favoring a correlation-based approach, and the redundancy of SVM-derived results with those already reported, we have decided not to include these additional analyses in the revised manuscript. However, it is encouraging to see that a correlation between reactivation and recall is also observed with a different method.

5) The authors state they study "memory replay". Replay is commonly used in animal literature to refer to sequential reactivation of firing patterns (as observed during learning) and many groups are currently striving to detect something similar in humans. The current work does not show sequential replay, it only shows that an indirect measure of memory reactivation is related to memory performance. They should thus refrain from using the term replay.

We appreciate the reviewers’ point that “replay” is often used to describe sequential reactivation in animal studies. There may be some debate in the field as to how this term should be used, and we chose to implement the term to describe a non-sequential measure of reactivation in fMRI, as has been done previously (e.g., Deuker et al., 2014). We accept, however, that this is not common practice. Therefore, we replaced the term “replay” with the term “reactivation” throughout the revised manuscript (including in the title). We clearly defined our terminology in the revised Introduction (first paragraph).

6) Why was the rigorous way to ensure no artefacts by head motion applied to sleep data not similarly applied to wake data where more head movements can be expected? Please justify.

We thank the reviewers for the shrewd observation that motion correction was applied more rigorously for the sleep data compared to the wake data. In our experiment, the sleep session consisted of one long (~75 min) fMRI run. Although we discarded volumes acquired prior to the first odor presentation and after the last odor presentation, the odor delivery period spanned much of the run, and on average, the sleep data consisted of ~950 volumes (range = 401-1416). Because we collected such a large number of fMRI volumes for the sleep session, there were sufficient degrees of freedom in our general linear model to incorporate 28 movement parameters (in addition to the other condition-specific parameters) without overfitting or otherwise compromising parameter estimates. In contrast, the wake data included a set of short 2.25 min runs, each including only 58 volumes. If we had included the full 28 movement parameters, our ability to reliably estimate parameters of interest would have been substantially reduced. This was the reason why we opted for a less rigorous motion correction algorithm for the wake data.

It is also worth noting that during the long, continuous sleep scan (> 1 hour), there were inevitable shifts in head position, so we felt it was prudent to apply more rigorous motion correction to the sleep data. The wake fMRI sessions, by comparison, were considerably shorter, and excessive head motion was highly unusual, so we felt that it was less important to apply rigorous motion correction to the wake data. We have added a short justification regarding these points in the Materials and methods section in the revised text (subsection “Multivoxel pattern analysis: re-emergence of category information during sleep”).

7) Why does the sleep analysis not follow the same rationale as the wake analysis, first subtracting a mean of both "categories" (here cue responses), then calculating pairwise correlations?

We regret not having explained this inconsistency in the original manuscript. While the wake data were effectively decorrelated (such that mean activity across the four categories was subtracted from each voxel prior to computing pairwise correlations), this same approach could not be applied to the sleep data. Because the sleep model only included two odor cues, if we had subtracted mean activity across the two cues, we would have created two anticorrelated vectors, essentially reducing our two data points to a single data point. We have amended the Materials and methods section to explain this inconsistency (subsection “Multivoxel pattern analysis: re-emergence of category information during sleep”).

Moreover, to mitigate any concerns that our findings were reliant on this discrepancy, we re-ran the main analysis without implementing the decorrelation step for the wake data. The outcome was effectively identical to our original findings.

8) Figure 2B. The behavioural reaction time measure might be confounded by accuracy. Were there differences in number of correct trials for interference learning for cued vs. non-cued items? Do reaction time results hold if only correct trials are analysed? If that difference exists and the reaction time results do not hold when only correct trials are analysed, please report this in the paper with a note of caution.

We thank the reviewers for this comment. Indeed, if recall accuracy differed across conditions, this could confound reaction time measurements. To ensure that this was not the case, we compared recall accuracy across cued and non-cued items. We found that there was no difference in recall accuracy across conditions (t_(17)_ = 0.04, *p* = 0.97). As such, recall accuracy cannot explain the difference in reaction times for cued versus non-cued objects, obviating the need to analyze correct trials in isolation. To make this transparent, we have included this result in the revised manuscript (subsection “Odor cues bias memory consolidation toward their associated object categories”, third paragraph).